# A Study of the Association between the Stringency of Covid-19 Government Measures and Depression in Older Adults across Europe and Israel

**DOI:** 10.3390/ijerph18158017

**Published:** 2021-07-29

**Authors:** Gina Voss, Andreia F. Paiva, Alice Delerue Matos

**Affiliations:** 1Communication and Society Research Centre, Institute of Social Sciences, University of Minho, 4710-057 Braga, Portugal; andreiafonsecapaiva@gmail.com (A.F.P.); adelerue@ics.uminho.pt (A.D.M.); 2Department of Sociology, Institute of Social Sciences, University of Minho, 4710-057 Braga, Portugal

**Keywords:** COVID-19, mental health, depression, stringency measures, older adults, SHARE

## Abstract

**Background**: The COVID-19 pandemic is having major adverse consequences for the mental health of individuals worldwide. Alongside the direct impact of the virus on individuals, government responses to tackling its spread, such as quarantine, lockdown, and physical distancing measures, have been found to have a profound impact on mental health. This is manifested in an increased prevalence of anxiety, depression, and sleep disturbances. As older adults are more vulnerable and severely affected by the pandemic, they may be at increased psychological risk when seeking to protect themselves from COVID-19. **Methods**: Our study aims to quantify the association between the stringency of measures and increased feelings of sadness/depression in a sample of 31,819 Europeans and Israelis aged 65 and above. We hypothesize that more stringent measures make it more likely that individuals will report increased feelings of sadness or depression. **Conclusions**: We found that more stringent measures across countries in Europe and Israel affect the mental health of older individuals. The prevalence of increased feelings of sadness/depression was higher in Southern European countries, where the measures were more stringent. We therefore recommend paying particular attention to the possible effects of pandemic control measures on the mental health of older people.

## 1. Introduction

The rapid spread of the SARS-CoV-2 virus (COVID-19) has had an impact on all aspects of society, with major adverse consequences for the mental health of individuals worldwide [1,2].

Lockdown subsequently became the most favoured global response to contain the pandemic [3]. The stringent measures imposed by governments—such as lockdown—which may differ between countries, commonly include closing schools and workplaces, cancelling public events, travel restrictions, stay at home requirements, quarantine, bans on public gatherings and contact tracing [4]. These restrictions have caused significant disruption globally and affected individuals, families, communities, and entire countries [5]. They significantly alter individuals’ routines, normal activities, and daily lives. Alongside the direct impact of the virus [6], these stringency measures have been found to have a profound effect on individuals’ mental health. There is a greater prevalence of anxiety, depression, and sleep disturbances [7,8] resulting in an increased mental health burden. The WHO has indicated its concern about the severe mental health and psychosocial consequences of the pandemic [9].

Several researchers have documented the psychological impacts of the pandemic. A study by Huang and Zhao [10] reported a high prevalence of general anxiety disorder and poor sleep quality in Chinese individuals during the outbreak of COVID-19. Lei et al. [11] observed a higher prevalence of anxiety and depression in individuals affected by the quarantine compared to those who were not affected by Chinese quarantine measures in early February 2020.

The consequences of the pandemic for mental health in Europe have also begun to be studied across different countries, but there are only a few studies examining the association with stringency measures. In Germany, regions with stricter restrictions experienced a significant increase in the number of helpline contacts on mental health issues [12]. Higher restrictions due to lockdown measures, a greater reduction in social contacts and greater perceived changes in life were associated with higher mental health impairments [8]. In Ireland, a study by Bailey et al. [13], involving 150 community-dwelling older adults, indicated a significant decline in self-reported mental health in individuals shielding from COVID-19, with 70 per cent of individuals reporting a low mood at least some of the time and 12 per cent reporting a low mood very often. A longitudinal study of Dutch older adults comparing emotional well-being before and during the pandemic showed that social loneliness and emotional loneliness increased during COVID-19 [14]. In a cross-European survey by Arpino et al. [15] conducted with 4,207 individuals aged 50 plus in three European countries (France, Spain, and Italy), the authors found that about 50 per cent of the sample reported feeling sad or depressed more often than usual during the lockdown. The ramifications of the COVID-19 pandemic and the environment surrounding individuals are likely to leave psychological scars, with experts agreeing that there has been a surge in mental health problems worldwide [16,17].

Older adults are more vulnerable and severely affected by the pandemic. This group of individuals has a higher rate of morbidity and mortality compared to younger groups [18,19,20]. Their daily social activities, such as walking and talking outside, meeting with friends, voluntary activities, and gatherings, have been restricted due to quarantine orders and transport controls. This has exacerbated the challenges of the pandemic and has had a severe impact on their mental health [21,22]. Lockdown measures are a “perfect storm” for mental illness in older individuals [23]. These individuals have been socially isolating since the beginning of the pandemic, and this has increased their sense of loneliness and social isolation. They are therefore more prone to developing severe mental and physical health issues [21]. Isolation from society makes it more likely that older adults will experience feelings of loneliness and other physical problems, which in turn foster a cascade of mental health problems [24]. This vulnerable group is at increased psychological risk, as individuals cocooning/shielding [25] from COVID-19 report worse mental health [9].

People who experienced feelings of sadness or depression before the pandemic have a higher risk of being negatively affected [16]. The disruption of daily activities, physical and social isolation, restrictions on movement and other measures imposed by governments to contain the spread of the virus are likely to trigger a stress response that affects the mental well-being of individuals, especially those with pre-existing vulnerability [16,26,27]. Mental health and well-being outcomes are an important indicator of future health consequences of the pandemic and will influence and drive many other individual choices, behaviours, and outcomes [1]. Inspired by the vulnerability stress model [26], this study therefore aims to quantify the association between the stringency measures and increased feelings of sadness or depression. We hypothesize that the greater the stressor (stringency measures), the more likely it is that individuals will report increased feelings of sadness or depression, while also controlling for demographic and socioeconomic variables usually associated with depression and individual stressors such as closeness of contact with COVID-19.

## 2. Materials and Methods

### 2.1. Study Population

This study uses data from the Survey of Health, Ageing and Retirement in Europe (SHARE), SHARE COVID-19, wave 8, preliminary release 0.0.1. beta [28]. SHARE is a multidisciplinary, cross-national panel survey that provides data on the health, socioeconomic status, and social and family networks of people who are 50 years and older. However, the appearance of COVID-19 interrupted the data collection for wave 8 (about 70 per cent complete) and the SHARE-COVID-19 questionnaire was developed. It covers the same topics as the regular SHARE questionnaire but is considerably shorter and targeted to the COVID-19 living situation. It was collected via computer-assisted telephone interviews (CATIs) from June to August 2020 in a total of 27 SHARE countries (Germany, Sweden, Netherlands, Spain, Italy, France, Denmark, Greece, Switzerland, Belgium, Israel, Czech Republic, Poland, Luxembourg, Hungary, Portugal, Slovenia, Estonia, Croatia, Lithuania, Bulgaria, Cyprus, Finland, Latvia, Malta, Romania, and Slovakia).

Additional methodological details are available elsewhere [28,29,30]. In our study, the sample was restricted to respondents aged 65 plus. Malta was excluded from our sample due to the lack of observations in our interest variable, and the Netherlands was also excluded due to the lack of observations in some variables. The final sample size for this study was 31,819 individuals.

The SHARE project, coordinated internationally by the Max Planck Institute for Social Law and Social Policy (Germany), has been approved by the Ethics Council of the Max Planck Society for the Advancement of Science and by the ethics committees of the institutions responsible for the study in the participating countries.

### 2.2. Outcome Variable

Mental health is the outcome variable and is based on a subjective evaluation of feelings of sadness or depression during the COVID-19 pandemic. All respondents answered two questions: (1) “In the last month, have you been sad or depressed?” and (2) “Has that been more so, less so, or about the same as before the outbreak of Corona?”. Individuals were classified into two groups: as presenting “increased sadness/depression” if they answered “Yes” to question (1) and “More so” to question (2). Those who answered “No” to question (1) were classified as “no sadness/depression”. Respondents who answered “less so” or “about the same” to the second question were deleted from our sample. This study therefore comprises individuals who reported an increase in sadness or depression in the month preceding the interview conducted during the pandemic, and those who stated that they had not felt sadness or depressed during the same reference period.

### 2.3. Interest Variable

The independent and primary interest variable of our study is the COVID-19 Government Response Stringency Index from the Oxford Coronavirus Government Response Tracker (OxCGRT) [4]. This index is constituted by nine measures of day-to-day variation in governments’ responses to tackling the pandemic (closings of schools and universities, closings of workplaces, cancelling public events, limits on gatherings, closing of public transport, orders to “shelter-in-place” and otherwise confine to the home, restrictions on internal movement between cities/regions, restrictions on international travel, presence of public info campaigns). This index is a score from 0 to 100, with a higher score indicating a more stringent response. As the index reflects daily values and our goal is to see the effect of restrictions over a period of time (March 11th—confirmation of the world pandemic by WHO [31] to August 14th—the end of the SHARE-COVID-19 interviews [28]), we calculated the mean score of the index for that period, per country.

### 2.4. Confounders

The confounders in our study were selected based on the literature review. Age of the respondent in 2020 and sex were selected as control variables. To consider the respondents’ socioeconomic position, two indicators were used: education and financial distress. Education was measured according to the highest level of education attained using the standardized coding of the International Standard Classification of Education (ISCED-97). This variable was categorized into three groups: low educational level (ISCED-97 levels 0–2 corresponding to lower secondary school at the most); medium educational level (ISCED-97 level 3, upper secondary school) and high educational level (ISCED-97 levels 4-6 corresponding to post-secondary school). As the question about level of education was not used in SHARE-COVID-19, the information was therefore imputed from wave 8 (release 0.0.0), wave 7 (release 7.1.1) [32] or wave 6 (release 7.0.1) [33]. As income is not covered in SHARE-COVID-19 and is an indicator that can vary substantially in a short period of time—which discourages the use of information reported in previous waves—we used a proxy indicator of income, which is available in SHARE-COVID-19: financial distress. This is assessed by the question “Thinking of your household’s total monthly income since the outbreak of Corona, would you say that your household is able to make ends meet?”. The answers were reclassified into two groups: ‘with great difficulty’/‘with some difficulty’ as “Yes”, and “fairly easily’/‘easily’ as “No”. Self-reported health was also used in the model and was accessed by the question “How was your health before the outbreak”. The answers ranged from 1 to 5 (excellent, very good, good, fair and poor).

Finally, we created a variable related to closeness of contact with COVID-19 with the questions: “Since the outbreak of Corona, have you or anyone close to you experience symptoms that you would attribute to the Covid illness, e.g., cough, fever, or difficulty breathing?”; “Have you or anyone close to you been tested for the Coronavirus and the result was positive, meaning that the person had the Covid disease?”; “Have you or anyone close to you been hospitalized due to an infection from the Coronavirus?”; “Has anyone close to you died due to an infection from the Coronavirus? “. All questions were dichotomous (yes or no), and individuals were classified as “No” if they answered no to all questions and “Yes” if they answered yes to at least one of the questions above.

### 2.5. Statistical Analyses

This study was carried out in two stages. Firstly, to characterize our study population, univariate descriptive statistics were applied using calibrated individual weights, as the SHARE survey does not have a uniform sample design. To analyse whether the sociodemographic variables and the stringency measures index showed significant differences between the increased sadness/depression group and the no sadness/depression group, tests for a two-group comparison (*T*-tests (*t*) and chi-square ($\chi^{2}$) tests) were performed. Effect size measures (Cohen’s d/Phi) were used to complement these analyses.

Secondly, to examine the association between the stringency measures index and mental health, a logistic regression was performed. The need for multilevel models was tested by calculating the Intraclass Correlation Coefficient (ICC) of the null model to determine the extent to which the variance in mental health was explained by country differences. The Intraclass Correlation Coefficient (ICC) of the null model was 4.3 per cent, lower than the recommended cut-point of 5 per cent. For this reason, there was no need to use multilevel modelling [34]. The logistic regression model was adjusted for age, sex, education, financial distress, self-reported health, closeness of contact with COVID-19 and our interest variable, the stringency measures index. Odds ratios (OR), 95 per cent confidence intervals (CIs), and significance (where *p*-values of < 0.05 were considered statistically significant) are presented in the table. Statistical analyses were conducted using R software, version 4.0.2, and IBM SPSS Statistics 27.

## 3. Results

In our sample, the mean age of participants is 75.3 years (SD = 7.5) and women constitute 55.4 per cent of our sample. In addition, 43.0 per cent had completed primary education or less, 33.7 per cent had completed secondary education and 23.3 per cent had completed post-secondary education. Furthermore, across the study sample, 28.7 per cent of the respondents reported being financially distressed and 47.8 per cent reported feeling good about their general health. Altogether 84.6 per cent reported never having had contact with COVID-19 disease (closeness to COVID-19 variable), and 21.9 per cent reported an increase in sadness/depression in the month before the interview conducted during the pandemic.

Characteristics of the study groups (increased sadness/depression group and no sadness/depression group) are displayed in Table 1. Without controlling for confounders, all the variables listed in Table 1 differed statistically in the two groups, although with no significant effect size.

Figure 1 shows the prevalence of increased feelings of sadness or depression and the mean scores of the stringency measures index by country. Overall, the highest prevalence of increased sadness/depression was reported in Portugal, Italy, and Spain (39.4; 30.8; 28.2 per cent, respectively), while the lowest was reported in Denmark, Slovenia, and Czech Republic (10.3; 10.6; 10.8 per cent, respectively). Countries with a higher mean level on the stringency measures index at the time of interview were Israel, Portugal, Cyprus, and France, while Estonia, Finland and Luxembourg reported a lower mean level on the stringency measures index.

Our logistic regression model shows that, regarding the association between increased sadness/depression and the stringency measures index, on average, for each extra point on the index, the chances of an individual having increased sadness/depression rose by 3 per cent (OR = 1.03 (1.03; 1.04)). This means that more stringent measures were statistically associated with worsened mental health (Table 2).

We did not find any statistical significance between age and increased feelings of sadness/depression, meaning that increased age did not make increased feelings of sadness/depression more likely in our study population (aged 65 and plus). Our results also revealed that women had a 117 per cent greater likelihood than men of having increased feelings of sadness/depression (OR = 2.17 (2.03; 2.31)). When compared to individuals with primary education or less, respondents with secondary education and post-secondary education were less likely to have increased feelings of sadness/depression (OR = 0.87 (0.80; 0.93); OR = 0.92 (0.85; 1.00), respectively). Individuals who reported being financially distressed were more likely to report an increased feeling of sadness or depression (OR = 1.34 (1.25; 1.43)). This is also the case for individuals who reported closeness of contact with COVID-19, as they were 59 per cent more likely to have increased sadness/depression (OR = 1.59 [1.46; 1.73]). Regarding self-reported health, individuals with worse levels in physical health were also more likely to report feelings of increased sadness/depression (OR =1.77 (1.71; 1.84)).

## 4. Discussion

Our results showed that, for each additional level of stringency measures index, individuals were 3 per cent more likely to report increased feelings of sadness or depression, when controlling for demographic and socioeconomic variables usually associated with depression, and for individual stressors such as closeness of contact with COVID-19 disease. These results can be interpreted based on the vulnerability stress model according to which extrinsic psychosocial stressors, such as a pandemic crisis and all the associated challenges, play a major role in psychological distress. Mental illness will manifest itself when an individual can no longer tolerate stress because their stress threshold is exceeded [16,26,27]. The results support our hypothesis based on the vulnerability stress model that suggests that the more severe the measures to contain the epidemic, the more likely it is that individuals will report increased feelings of sadness or depression.

The above results are consistent with the outcomes of the German study by Benke et al. [8] of adults aged from 18 to 95 years. The authors found that higher distress related to the restriction of social contacts was associated with increased mental health impairments, mainly depression and anxiety. Although the studies did not control for the eventual effect of other variables associated with the mental health of individuals, they are noteworthy for having also found an association between the restrictive measures adopted during the pandemic and individual mental health. Indeed, Armbruster et al. [12] found that, during the first week of lockdown in Germany, there was an increased demand for psychological counselling and a spike in helpline contacts to deal with mental health struggles such as depression, loneliness, and fear. The authors also stated that the average effect was more pronounced in states that implemented stricter measures. A recent survey by Arpino et al. [15] in France, Spain, and Italy found that about 50 per cent of the sample, aged 50+, reported feeling sad or depressed more often than usual during the lockdown. In the same way, Krendl et al. [35] found that older adults in Indiana (EUA) reported higher depression after the onset of the pandemic while “Sheltering in Place” due to COVID-19. This was also the case in previous studies from past epidemics, such as the SARS outbreak, which revealed that individuals affected by quarantine reported higher levels of mental health struggles [21,36,37]. By quantifying the relationship between an index of stringency measures and increased feelings of sadness or depression in older adults, after controlling for other variables associated with their mental health, we contribute valuable information about mental health problems during the COVID-19 pandemic in European countries and Israel where stringency measures were adopted to contain the spread of the virus. It is possible that unfavourable feelings towards the pandemic and stringency measures can act as an extrinsic stressor and have an impact not only in the immediate present, but might increase the risk of developing psychological or psychopathological problems in the future [8,36].

Our study population is older adults (aged 65 plus), a vulnerable group that has been greatly affected by the pandemic, not only in terms of disease symptoms and even mortality, but also mental health due to possible disruptions to their day-to-day lives.

The results obtained from a gender analysis are in line with the previous literature [7,38]. This showed that women were more likely to have increased feelings of sadness/depression during the first wave of the COVID-19 pandemic and during lockdown measures [39,40]. A study by Banks and Xu [1] exploring the effects of the COVID-19 lockdown and social distancing in the UK revealed that the effects on mental health were significant and were greater for women. Benke [8] also found a greater association between increased depression and being a woman during the COVID-19 pandemic. We hypothesize that these increased feelings of sadness/depression may be associated with stress and, according to Sareen et al. [41], women are more prone to stress.

Regarding the role of education in predicting increased feelings of sadness/depression, our results are consistent with studies showing that older people with low socioeconomic status, mainly due to education or low income, are particularly at risk of developing depression [42,43] and that people with lower educational levels are also less likely to have support from mental care services [44].

Financial difficulties have been reported to be a stressor for individuals under quarantine orders. Our results are in line with another study which reports that individuals who experienced financial loss during quarantine due to the SARS outbreak in Canada were affected by psychological distress [45]. In our study, as our individuals were aged 65 plus, most individuals no longer work, but might still have financial struggles, as they may suffer financial losses regarding other means of making money, such as rent or pensions and also from the inability to pay for, or access, services that facilitate daily living during a pandemic. Previous research has suggested that the risk of mental health problems is exacerbated among the poorest older individuals, particularly those who are self-isolating. This is because they are the most reliant on social care and community support and are also less likely to be technically literate or have access to the technological means that enable remote social contact with family and friends [24].

Previous studies have shown that one way to improve mental health during the COVID-19 pandemic is taking part in physical exercise, reducing one’s consumption of COVID-19 related information (such as social media and news outlets) and implementing remote mental health care [46]. Our results showed that individuals with lower levels of self-reported health are also more likely to report feelings of increased sadness/depression. In another study, the authors [47] found that higher levels of physical activity in older adults, while still maintaining the physical distancing measures, helped to alleviate some of the negative mental health symptoms.

Regarding the individual stressor—closeness of contact with COVID-19—our results are also in line with recent research reporting that individuals with coronavirus symptoms and diagnosis also reported a deterioration in mental health [1] and an increased feeling of depression [8]. This included individuals who had a close relative who was infected [7].

When governments need to apply stringency measures, it is recommended that they also implement mitigation measures. Thus far, the main mitigation measures implemented by governments have focused on emergency containment of the virus and measures of a financial nature. However, the effects of the pandemic and consequently of the restrictions imposed can be very harmful to mental health. We recommend implementing protection policies to minimize the impact on mental health, with a specific focus on older individuals, especially women and less educated individuals. These should include improving support in the community and social services, as well as stimulating the use of technology to promote social connections to prevent social isolation and loneliness, which can increase the probability of depression in this population [6,36,48,49,50].

### Limitations and Strengths

The findings of this research need to be interpreted within a framework that is sensitive to the limitations of the study. As this study has a cross-sectional design, we cannot assume causality. Furthermore, the data collected refer to a specific time point in the pandemic (first wave).

It is of interest to continue this research in a longitudinal framework to test our hypothesis that the stringency measures during the pandemic have an impact on mental health in the long term. This analysis will be possible with the data from future SHARE waves.

This study also has strengths, one of the main ones being the use of the COVID-19 Government Response Stringency Index from the Oxford Coronavirus Government Response Tracker (OxCGRT) [4]. This collects information on nine restrictive measures and reflects more robustly the degree of restriction of the stringency measures implemented across countries and the possibility of quantifying the association between these and increased feelings of sadness or depression in European countries and Israel. Another strength of our study is the fact that the association between the stringency measures index (an extrinsic stressor) and increased feelings of sadness or depression was controlled for an individual stressor, closeness of contact with COVID-19, in addition to several demographic and socioeconomic variables, which can also be related to feelings of sadness or depression. It therefore contributes much greater precision to the model and hence to the reported values.

## 5. Conclusions

To sum up, we found that higher levels of stringency measures across countries in Europe and Israel affect the mental health of older individuals. There was a greater prevalence of increased feelings of sadness/depression in Southern European countries (Portugal, Italy, and Spain) and these countries also implemented stricter measures than other countries.

Although stringency measures, such as physical distancing, workplace closures and travel restrictions, are necessary to prevent and contain the spread of the virus, these measures are also associated with increased psychological distress. Since these measures are essential during pandemics, they need to be handled with caution. As seen in a review of the studies, making these stringency measures easier and more tolerable for individuals can be achieved by providing people with careful and clear information, providing health-oriented activities to occupy time, and ensuring that all individuals have enough supplies for their basic needs [36]. Other options for tackling feelings of depression in older adults include medication or counselling, as well as behavioural strategies that promote self-help, positive emotions, and feelings, assisted by trained professionals, in addition to public health messaging that contributes to preventing long term mental health problems among individuals who present vulnerabilities [23]. Health and government officials should also make mental health a priority in times of crisis by improving the monitoring and reporting of psychological distress in individuals and arrange mechanisms to improve access to mental health care, with digital and non-digital interventions [6,46]. It is important to make sure that mental health issues are not suppressed and abandoned [51], since the psychological effects of particular actions to prevent the spread of the disease can have serious effects on the mental well-being of individuals [36], especially those individuals with pre-existing mental health struggles, as they are more likely to experience more severe psychological consequences [16,52].

## Figures and Tables

**Figure 1 ijerph-18-08017-f001:**
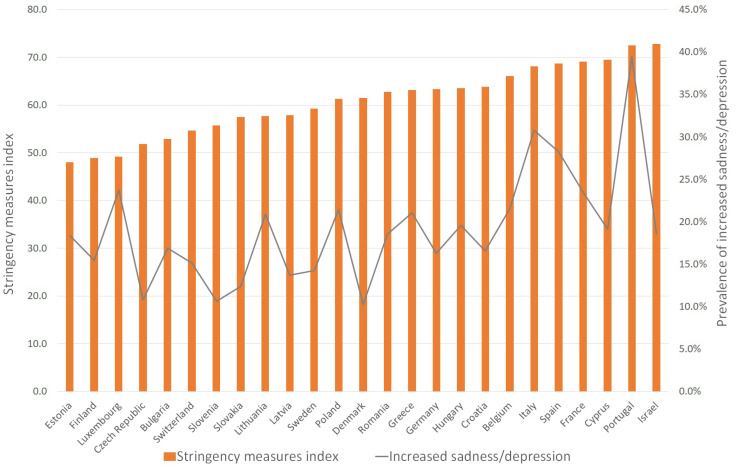
Prevalence of increased sadness/depression by country. Source: Preliminary SHARE COVID-19, wave 8, release 0.0.1 beta. Conclusions are preliminary. Weighted data. Notes: The sample was limited to individuals aged 65+.

**Table 1 ijerph-18-08017-t001:** Characteristics of the no sadness/depression group and the increased sadness/depression group.

	No Sadness/Depression	Increased Sadness/Depression				
	(N = 25,879)	(N = 5940)	*t*/$\chi^{2}$	*p*-Value	Cohen’s d/Phi	CI 95 Per Cent
Age, mean (SD)	75.01 (7.46)	76.40 (7.67)	−11.363	0.000 ***	−0.19	−0.214–−0.157
Sex						
Female (per cent)	51.20	70.42	646.291	0.000 ***	0.14	0.132–0.153
Male (per cent)	48.80	29.58				
Education						
Primary or less (per cent)	39.99	53.59	274.203	0.000 ***	0.09	0.082–0.104
Secondary (per cent)	35.49	27.37				
Post-secondary (per cent)	24.52	19.04				
Financial distress						
No (per cent)	74.09	61.35	322.387	0.000 ***	0.10	0.090–0.112
Yes (per cent)	25.91	38.65				
Self-reported health, mean (SD)	3.07 (0.89)	3.58 (0.89)	−36.334	0.000 ***	−0.57	−0.602–−0.545
Closeness of contact						
with COVID-19						
No (per cent)	85.20	82.43	65.387	0.000 ***	0.05	0.034–0.056
Yes (per cent)	14.80	17.57				

Source: Preliminary SHARE COVID-19, wave 8, release 0.0.1 beta. Conclusions are preliminary. Weighted data, N = 31,891. Notes: *t*/$\chi^{2}$ (*t*-test and chi-squared test), CI (confidence intervals). Tests for effect size: Cohen’s d: small effect (≥0.20); medium effect (≥0.50); large effect (≥0.80); Phi: small effect (≥0.10); medium effect (≥0.30); large effect (≥0.50). *** Significant associations (*p* < 0.05). The sample was limited to individuals aged 65+.

**Table 2 ijerph-18-08017-t002:** Logistic regression for increased feelings of sadness/depression.

	OR (CI 95 Per Cent)
(Intercept)	0.00 (0.001–0.003) ⋆⋆⋆
Age (years)	1.00 (1.00–1.01)
Female	2.17 (2.03–2.31) ⋆⋆⋆
Education	
Primary or less	ref.
Secondary	0.87 (0.80–0.93) ⋆⋆⋆
Post-secondary	0.92 (0.85–1.00) ⋆
Financial distress	1.34 (1.25–1.43) ⋆⋆⋆
Self-reported health	1.77 (1.71–1.84) ⋆⋆⋆
Closeness of contact with COVID-19	1.59 (1.46–1.73) ⋆⋆⋆
Stringency index	1.03 (1.03–1.04) ⋆⋆⋆

Source: Preliminary SHARE, wave 8, release 0.0.1 beta. Conclusions are preliminary. Notes: *Ref* reference group, *OR* odds ratio, *CI* confidence intervals. Significant associations: ´⋆⋆⋆´ < 0.001; ’⋆⋆’ < 0.01; ’⋆’ < 0.05. The sample was limited to individuals aged 65+.

## Data Availability

SHARE data available through individual user registration. All details about the application and registration process can be found at http://www.share-project.org/data-access/user-registration.html accessed on 29 June 2021.

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
