# Peer review of "A Study of the Association between the Stringency of Covid-19 Government Measures and Depression in Older Adults across Europe and Israel"

_ijerph, 2021, doi:10.3390/ijerph18158017_

Round 1
Reviewer 1 Report
The manuscript “A study of the association between the stringency of Covid-19 government measures and depression in older adults across Europe and Israel” perfectly fits the scope of the special issue.
I find this paper very interesting for several reasons: it is based on the most recent wave of data from the SHARE European survey (SHARE COVID-19, wave 8, preliminary release 0.0.1.beta) with citizens 65 years or older, from representative samples across Europe and using new and reliable indicators as the Oxford Coronavirus Government Response Tracker (OxCGRT). This is a quite comprehensive measure, considering the day-to-day variation in governments’ responses to tackling the pandemic e.g., travel restrictions, school closures, limits on gatherings, and stay-at-home requirements. The index ranges from 0 to 100, with a higher score indicating a more stringent response.
The design and the variables are adequately selected. The logistic regression model shows that, regarding the association between increased sadness/depression and the stringency measures index, there is evidence that more stringent measures were statistically associated with worsened mental health when controlling by key variables as health, education, ...
I only would like more information in the discussion section regarding two issues. First, when concluding "unfavorable feelings towards the pandemic and stringency measures can act as an extrinsic stressor, and have an impact, not only in the immediate present but might increase the risk of developing psychological or psychopathological problems in the future" as this is based in two former studies, in previous research but obviously current research needs in the future to have a follow-up with a longitudinal perspective. So, could the authors comment on this possibility? Second, when discussing about financial difficulties, indeed they have been reported to be a stressor for individuals under quarantine orders. An additionally we know by index OxCGRT, as clearly seen in Figure 1 their strength of stringency measures in some southern countries. In discussion are omitted cultural issues as in these type of countries older people traditionally "support" the family economy meanwhile in other countries their members could live more independently. So, could exist additional stressors, additional problems due to cultural issues, that could be discussed.
I totally agree with the conclusions as the authors recommend implementing protection policies to minimize the impact on mental health, with a specific focus on older individuals, especially women and less educated individuals..
An important limitation of the work is already mentioned, it only considers the responses and the situation of a short time during the pandemic, so, different countries could live different situations at this time. The only recommendation is to continue further research in a longitudinal design.
Author Response
Title: A study of the association between the stringency of Covid-19 government measures and depression in older adults across Europe and Israel
Authors: Gina Voss, Andreia F. Paiva, Alice Delerue Matos
Author’s response to reviews:
Dear Reviewer,
We thank you for providing thoughtful feedback on our manuscript (ijerph-1298502), and for giving us an opportunity to improve it. Below, we have addressed each of your comments on a point-by-point basis and have revised the manuscript accordingly.
Reviewer 1
Reviewer comment: I only would like more information in the discussion section regarding two issues. First, when concluding "unfavorable feelings towards the pandemic and stringency measures can act as an extrinsic stressor, and have an impact, not only in the immediate present but might increase the risk of developing psychological or psychopathological problems in the future" as this is based in two former studies, in previous research but obviously current research needs in the future to have a follow-up with a longitudinal perspective. So, could the authors comment on this possibility?
Response: Thank you very much for raising this issue. We agree that when new SHARE data become available, the current research would benefit from being re-examined from a longitudinal perspective. Adopting this perspective will allow to test the relevance of our hypothesis. Thus, following your comment, we have added a note on this issue in the limits and strengths section of the manuscript (page 8, lines 357-360).
Reviewer comment: Second, when discussing about financial difficulties, indeed they have been reported to be a stressor for individuals under quarantine orders. An additionally we know by index OxCGRT, as clearly seen in Figure 1 their strength of stringency measures in some southern countries. In discussion are omitted cultural issues as in these type of countries older people traditionally "support" the family economy meanwhile in other countries their members could live more independently. So, could exist additional stressors, additional problems due to cultural issues, that could be discussed.
Response: Thank you for pointing this out. We are aware of its relevance so, in previous analyses, we looked at the question regarding financial support from family members in the SHARE-Covid-19 questionnaire. Nevertheless, we found that only 0.5 per cent of the total sample received financial help from family members, with only one country (Bulgaria) exceeding 2 per cent. This is the reason why we did not include this variable in our statistical model.
Reviewer comment: The only recommendation is to continue further research in a longitudinal design.
Response: Thank you for your recommendation. This is already in our future research plans.
Reviewer 2 Report
Well done analysis, interesting argument. Only just a minor suggestion: since the data derives from a probably personal data base and are shown in the table only Female and male, instead of the term gender the term sex would be more appropriate in table 1
Author Response
Title: A study of the association between the stringency of Covid-19 government measures and depression in older adults across Europe and Israel
Authors: Gina Voss, Andreia F. Paiva, Alice Delerue Matos
Author’s response to reviews:
Dear Reviewer,
We thank you for providing thoughtful feedback on our manuscript (ijerph-1298502), and for giving us an opportunity to improve it. Below, we have addressed each of your comments on a point-by-point basis and have revised the manuscript accordingly.
Reviewer 2
Reviewer comment: Only just a minor suggestion: since the data derives from a probably personal data base and are shown in the table only Female and male, instead of the term gender the term sex would be more appropriate in table 1.
Response: Thank you for your recommendation. We have replaced gender by sex in table 1 and in the text when we refer to this variable.
Reviewer 3 Report
General Comments:
The Covid-19 pandemic is having major adverse consequences for the mental health of individuals worldwide. This study aims to quantify the association between the stringency of measures and increased feelings of sadness/depression in a sample of 31819 Europeans and Israelis aged 65 and above. The result showed that more stringent measures across countries in Europe and Israel affect the mental health of older individuals and more stringent measures make it more likely that individuals will report increased feelings of sadness or depression. The paper is generally well-written and informative. However, there are some points that authors should be addressed to further convince the audience regarding their arguments.
- Introduction section: As the aim of this study was to quantify the association between strict measures and increased sadness or depression. Information that is not relevant to that purpose is redundant. It is recommended that this section should be simplified to make the structure and logical relationship of the article clearer for the audience.
- Method section:
- Did the authors compare the variability in the impact of the total number of infections and deaths on respondents' mood in each country?
- With regard to financial distress, have the authors conducted a survey on whether people over the age of 65 receive a pension or have to work to secure a living? Because the degree of impact on mental health varies between pensioners and non-pensioners, which affects the final statistical results of the study. Please clarify this situation.
- Results and Discussion section:
- The lockdown days are different in Europe and Israel. Did the authors analyze how the number of days of lockdown affected older people? The authors should explain the findings of the study in relation to the number of days of lock down for comparison.
- How did the authors quantify the mean scores of the stringency measures index by country? Please provide clarification in the methodology section.
Author Response
Title: A study of the association between the stringency of Covid-19 government measures and depression in older adults across Europe and Israel
Authors: Gina Voss, Andreia F. Paiva, Alice Delerue Matos
Author’s response to reviews:
Dear Reviewer,
We thank you for providing thoughtful feedback on our manuscript (ijerph-1298502), and for giving us an opportunity to improve it. Below, we have addressed each of your comments on a point-by-point basis and have revised the manuscript accordingly.
Reviewer 3
Reviewer comment: Introduction section: As the aim of this study was to quantify the association between strict measures and increased sadness or depression. Information that is not relevant to that purpose is redundant. It is recommended that this section should be simplified to make the structure and logical relationship of the article clearer for the audience.
Response: Thank you for your suggestion. We skip general statements in order to better focus the introduction on the main topic (stringency measures and its association with the mental health of older adults).
Reviewer comment: Method section: Did the authors compare the variability in the impact of the total number of infections and deaths on respondents' mood in each country?
Response: We considered the impact of the total number of infections and deaths on respondents' mood in each country. The variable used as confounder in our model to test this was closeness of contact with Covid-19. This variable contains information on infections on the individual and anyone close to the respondent, and information on mortality of anyone close to the individual, as explained in page 4, lines 174-182.
Reviewer comment: With regard to financial distress, have the authors conducted a survey on whether people over the age of 65 receive a pension or have to work to secure a living? Because the degree of impact on mental health varies between pensioners and non-pensioners, which affects the final statistical results of the study. Please clarify this situation.
Response: We are aware that the impact of financial distress on mental health between working and retired individuals is different. For this reason, we decided to restrict our sample to individuals 65 years and older. In our sample, 94.3 per cent of the individuals have already retired.
Reviewer comment: Results and Discussion section: The lockdown days are different in Europe and Israel. Did the authors analyze how the number of days of lockdown affected older people? The authors should explain the findings of the study in relation to the number of days of lock down for comparison.
Response: The number of days of lockdown is already controlled in our statistical model. Indeed, the COVID-19 Government Response Stringency Index integrates nine measures for controlling the spread of the virus, of which a measure to "shelter-in-place" and to confine to the home. To make this more explicit in the text, we have now mentioned all the nine measures included in the index. Please see page 3, lines 138-151.
Reviewer comment: How did the authors quantify the mean scores of the stringency measures index by country? Please provide clarification in the methodology section.
Response: Since it is a daily observations index, the mean score, for each country, was calculated for the period of 11th of March to 14th of August. We slightly changed the text to make it clearer (page 3, lines 150-151).
Round 2
Reviewer 3 Report
I have reread the last version ; it has been improved dramatically.